

# Insight into the dynamics of a long run-out mass movement using single-grain feldspar luminescence in the Pokhara valley, Nepal

Anna-Maartje de Boer[1], Wolfgang Schwanghart[2], Jürgen Mey[2], Basanta Raj. Adhikari[3], Tony Reimann[1,4]

[1]Soil Geography and Landscape group & Netherlands Centre for Luminescence Dating, Wageningen University & Research, Wageningen, the Netherlands
[2]Institute of Environmental Science and Geography, University of Potsdam, Potsdam, Germany
[3]Department of Civil Engineering, Institute of Engineering, Tribhuvan University, Lalitpur, Nepal
[4]Institute of Geography, University of Cologne, Cologne, Germany

10  *Correspondence to*: Anna-Maartje de Boer (anna-maartje.deboer@wur.nl)

**Abstract.** Mass movements play an important role in landscape evolution of high mountain areas such as the Himalayas. Yet, establishing numerical age control and reconstructing transport dynamics of past events is challenging. To fill this research gap, we bring Optically Stimulated Luminescence (OSL) dating to the test in an extremely challenging environment: the Pokhara Valley in Nepal. This is challenging for two reasons: i) the OSL sensitivity of quartz, typically the mineral of choice for dating sediments younger than 100 ka, is poor, and ii) highly rapid and turbid conditions during mass movement transport hamper sufficient OSL signal resetting prior to deposition, which eventually results in age overestimation. Here, we first assess the applicability of single-grain feldspar dating of medieval mass movement deposits catastrophically emplaced in the Pokhara Valley. Second, we exploit the poor bleaching mechanisms to get insight into the sediment dynamics of this paleo-mass movement through bleaching proxies. The Pokhara valley is a unique setting for our case-study, considering the availability of an extensive independent radiocarbon dataset as a geochronological benchmark.

Single-grain infrared stimulated luminescence signals were measured at 50°C (IRSL-50) and post-infrared infrared stimulated luminescence signals at 150°C (pIRIR-150). Our results show that the IRSL-50 signal is better bleached than the pIRIR-150 signal. A bootstrapped Minimum Age Model (bMAM) is applied to retrieve the youngest subpopulation to estimate the paleodose. However, burial ages calculated with this paleodose overestimate the radiocarbon ages by an average factor of ~23 (IRSL-50) and ~72 (pIRIR-150), showing that dating of the Pokhara Formation with a single-grain approach was not successful for most samples. Some samples, however, only slightly overestimate the true emplacement age and thus could be used for a rough age estimation. Large inheritances in combination with the scatter in the single-grain dose distributions show that the sediments have been transported under extremely limited light exposure prior to deposition, which is consistent with the highly turbid nature of the sediment laden flood and debris flows depositing the Pokhara gravels.

To investigate the sediment transport dynamics in more detail, we studied three bleaching proxies: the percentage of grains in saturation 2D0 criteria, the percentage of best-bleached grains (2σ range of bMAM-De) and the overdispersion (OD). None of the three bleaching proxies indicate a spatial relationship with run-out distance of the mass movement deposits. We



interpret this as evidence for the lack of bleaching during transport, which reflects the catastrophic nature of the event. While not providing reliable burial ages of the Pokhara mass movement deposits, single-grain feldspar dating can potentially be used

as an age range finder method. Our approach shows the potential of luminescence techniques to provide insights in sediment transport dynamics of extreme and rare mass movement events in mountainous region.

## 1 Introduction

Mass movements such as landslides and debris flows belong to the most fatal and destructive geomorphological events in high-mountain areas (Froude & Petley, 2018). Sourced on steep mountain flanks, they are capable of displacing large volumes of

sediment over hundreds of meters to several tens of kilometres (Hewitt, 1988; Shugar et al., 2021). As these mass movements propagate downstream, they put people, buildings, agricultural land and infrastructure at risk, and impact river systems with high sediment loads, thus leading to widespread aggradation, affecting river habitats and drinking water quality. The 6th assessment report of the IPCC (IPCC, 2022) indicates that many regions will face an increase in landslide activity in a future changing climate, but there is still low confidence about these predictions.

How landslide activity changes over time can be inferred from paleo-evidence (Gariano & Guzzetti, 2016). Several studies, for example, have compared ages of landslide deposits with paleoclimatic records (Borgatti & Soldati, 2010, and references therein). In rapidly eroding mountain areas, however, this approach is hampered due to the low preservation potential of the landslide deposits (Croissant et al., 2017). Moreover, diamictic sediments are often difficult to clearly attribute to mass movements as they resemble glacial deposits (Hewitt et al., 1999; Weidinger et al., 2014). To this end, there are

multiple challenges in constraining the age of landslide deposits (Panek, 2015; Zech et al., 2009). For example, there often is a lack of organic material for radiocarbon dating and even if available, radiocarbon ages should be seen as maximum ages as they may be affected by older organic material incorporated in the displaced sediment mass (Panek, 2015).

To tap other avenues for dating mass movements, several studies have examined the potential of luminescence techniques (Eriksson et al., 2000; Fuchs et al., 2010; Tatumi et al., 2003). On geological timescales, mass movement deposits

are freshly eroded from the bedrock. For example, deep-seated mass movements in young and active orogens usually entrain material with a short, if any, sedimentary history. This entails that quartz is often less sensitive than feldspar (Duller, 2006; Preusser et al., 2006) and that the quartz signal sensitivity varies widely over sedimentary settings (Preusser et al., 2009). In mountainous areas, such as the Himalayas, freshly eroded quartz often shows very poor luminescence sensitivity (Chamberlain et al., 2017; Jaiswal et al., 2019; Scherler et al., 2015). In contrast, feldspar from plutonic or metamorphic bedrock bears a

measurable luminescence signal and has a higher intrinsic brightness than quartz (Guralnik et al., 2015). Moreover, the relatively short travel distance of mass movements and thus limited exposure time of individual grains often leads to heterogeneously bleached samples.

The degree to which samples are bleached is, aside from external factors, dependent on the intrinsic mineralogical properties as well as the signal used. Especially infra-red stimulated luminescence signals from feldspars are prone to poor



bleaching (Reimann et al., 2012) as the exposure to daylight resets the luminescence signal in feldspar at slower rates than in quartz (Godfrey-Smith et al., 1988; Thomsen et al., 2008), particularly with respect to the post-IR IRSL (pIRIR) signal (e.g., Buylaert et al., 2012; Kars et al., 2014). The challenge of heterogeneously bleached samples may be overcome by the application of single-grain measurement techniques (e.g., Bonnet et al., 2019). Single-grain analysis identifies best-bleached and heterogeneously bleached grains within samples (see review by Duller, 2008). For heterogeneously bleached samples –

insufficient light exposure to reset the luminescence signal - single-grain dating could systematically curtail age overestimation by distinguishing those grains which capture the "true burial age" from those which inherit a heterogeneously bleached signal. Statistical models, like the Minimum Age Model (MAM) demonstrate that single-grain pIRIR dating can be decisive in overcoming age overestimation (Brill et al., 2018).

Feldspar minerals are prone to anomalous fading, especially when using the conventional IRSL signal, which is a

major drawback in using feldspar for dating sediments. Anomalous fading is athermal loss of trapped charge resulting in dose underestimation. It is presumably caused by quantum mechanical tunnelling from the ground state of the trap (e.g., Jain & Ankjærgaard, 2011, and references therein). The combination of pIRIR stimulation, especially at moderate temperatures (<200 °C), and single-grain equivalent dose (De) analyses can correct for age underestimation effects caused by anomalous fading on burial age determination (Li et al., 2014; Reimann et al., 2012; Roberts, 2012; Thomsen et al., 2008), and potentially offer

opportunities in dating mass movement or mega-lake deposits (Zhang et al., 2022). Furthermore, recent methodological developments have shown the potential of single-grain feldspar luminescence beyond dating deposits. For example, heterogeneous bleaching can be used to unravel recent and past sediment dynamics and is a promising proxy in the fields of sediment tracing, sediment provenance, and sediment dynamics reconstruction (Bonnet et al., 2019; Chamberlain & Wallinga, 2019; Gray et al., 2019; Guyez et al., 2022; Reimann et al., 2015; Rhodes et al., 2022; Riedesel et al., 2018; Sawakuchi et al.,

2020). Another novel proxy that has been shown to be useful for sediment provenance analysis is the sensitivity of the quartz OSL signal (Souza et al., 2023).

In this study, we bring OSL dating to the test for sediment samples from the Pokhara Valley in central Nepal. The valley hosts extensive deposits of medieval mass-wasting events sourced from the Annapurna Region (Schwanghart et al., 2016; Stolle et al., 2017). We apply single-grain feldspar pIRIR dating to study the suitability of the method to determine the

age of catastrophic deposits emplaced by mass movement, thus expanding on few previous works that have attempted to obtain luminescence burial data for deposits in and around the Himalayan Mountain range (Hu et al., 2015; Lavé et al., 2022). The obtained burial ages are compared with an extensive dataset of radiocarbon ages which offers an independent benchmark (Yamanaka, 1982; Fort, 1987; Schwanghart et al., 2016). Moreover, we analyse three luminescence-based bleaching proxies and test their consistency with the notion of a rapid, long-run-out mass movement: the overdispersion, the percentage of

saturated grains and the percentage of best-bleached grains as identified by the bootstrapped minimum age model (bMAM).





## 2 Study site

We conducted fieldwork in the Pokhara Valley, Nepal (28.2°N, 83.0°E) (Fig. 1). The valley contains one of the world's steepest topographic gradients as elevations drop from the >7000 m high peaks of the Annapurna Massif down to Pokhara City (~800 m) in less than 35 km horizontal distance. Together with high seismicity in the region, this topographic setting primes the valley for very large, long-runout mass movements as testified by extensive mass-wasting deposits lining the valley bottom over a distance of ~70 km (Fort, 1984; Fort, 1987; Fort, 2009; Yamanaka, 1982). Among the deposited layers, the Pokhara Formation is the youngest one (Fig. 1a). The 100 m thick formation comprises several, up to 10 m thick beds of massive, matrix-supported, very poorly sorted, and locally fining-upward conglomerates, indicative of repeated surges of debris flows. The conglomerates are overlain by up to 40 m thick massive, clast-supported pebble- to granule-bearing sheets (Fig. 2). These sediments are interpreted to be deposited by rapid aggradation from turbulent, sediment-laden flow due to absence of current structures and erosive contacts. Dark Nilgiri limestone as well as kyanite-sillimanite gneiss, pyroxenic marble and augen gneiss dominate the lithological composition of the deposits and attest their High-Himalayan and Tethyan origin. The lithological composition further enables to distinguish the deposits from local sediments derived from Lesser Himalayan low-grade phyllitic bedrock (Schwanghart et al., 2016; Stolle et al., 2017). The estimated volume of 4-5 km³ of the Pokhara Formation can thus be unambiguously associated with a localized source in the Sabche Cirque (Lavé et al., 2022), which enables to infer transport distances.

Deposits of the Pokhara Formation form an extensive debris fan perched on and spatially confined by the Pokhara Valley (Fig. 2 a-c). The sediments are accessible downstream from the apex of this fan. Further upstream, in the Seti Gorge, the sediments are preserved in a few isolated hills within the valley (Lavé et al., 2022) which were not accessible at the time of our fieldwork. A radiocarbon-based geochronology of 37 radiocarbon ages from charcoal, wood, humic silt, peat, intact leaves, and soils (Fort, 1987; Schwanghart et al., 2016; Stolle et al., 2017; Yamanaka, 1982) was mostly recovered upstream in tributaries to the Seti River. The debris flow and flood deposits were swept up 7 km into these tributaries where reduction of flow velocities led to the deposition of fine materials (e.g., slackwater deposits) together with organic material often found at the top of silty beds. The (re-)calibrated (Ramsey, 2009) radiocarbon dates show that the emplacement of the Pokhara Formation is consistent with the timing of M>8 medieval earthquakes shattering the region between 1100 and 1344 C.E., with the 1222 C.E. earthquake being the event releasing most of the material from the Annapurna Massif (Stolle et al., 2017).

## 3 Materials and methods

### 3.1 Luminescence sampling and measuring

We collected samples along the entire flow distance between the apex and the most distal parts of the Pokhara Formation (Fig. 1 and 2a-c) for luminescence dating in autumn of 2019. Sample preparation and analysis were conducted at the Netherlands Centre for Luminescence dating (NCL) at Wageningen University under amber light conditions. The outer three centimetres



of sand from each tube were used for environmental dose rate (Dr) measurements and the rest of the sample was sieved at a grainsize of 212-250 µm for paleodose estimation. We added a 10% HCl⁻ solution to the 212-250 µm fraction to remove carbonates. Consecutively, organic matter (OM) was removed with a 10% $H_2O_2$ solution. To obtain K-feldspars with a density

ranging between 2.50 and 2.56 g/cm³, we conducted a heavy density separation of quartz and feldspar. First, quartz grains were removed by adding a liquid with a density of 2.63 g/cm³, and etched in a 10% HF⁻ solution. Subsequently, the K-feldspars were isolated from the Na-feldspars by adding a liquid with a density of 2.58 g/cm³, and mounted on single-grain discs. The bulk sediment for Dr measurement was dried overnight at 105°C and then weighted to calculate the moisture content by Loss On Ignition (LOI) (Heiri et al., 2001). Then the samples were ashed overnight at 550°C to determine their organic matter (OM)

content and grinded to obtain a homogeneous sample with particles smaller than 300 µm. Then, pucks were formed by adding melted wax and the activity of radioactive nuclides was measured with gamma spectrometry.

All luminescence measurements were performed on an automated Risø TL/OSL reader (DA 15). The samples were irradiated by a ⁹⁰Sr/⁹⁰Y beta-source with a dose rate of 0.098 Gy/s for all single-grain measurements. The infra-red stimulated luminescence emitted by the feldspar grains was detected through an I-410 filter to capture the typical K-feldspar blue emission

at 410 nm for the IRSL (at 50°C) and pIRIR (at 150°C) signal. In addition to the SAR measurement of equivalent doses, fading and dose recovery tests are performed (Table 1). For the dose recovery test sample, Pk-X18 was bleached for 48 hours in a solar simulator, subsequently dosed to 15 Gray and measured using the pIRIR-150 SAR protocol (Table 1, column on the left). The laboratory fading test was performed on multi-grain discs using the approach outlined in Auclair et al. (2003) on samples Pk-A02, Pk-B04, Pk-D15, and Pk-E16 (Table 1, column in the middle). To compare the OSL and radiocarbon data, an

overestimation was calculated in the following manner: the median value of the radiocarbon dataset (Schwanghart et al., 2016) was used as benchmark value and expressed in CE to calculate the ratio between a OSL burial age and the radiocarbon benchmark value. The newly obtained radiocarbon samples were calibrated with the IntCal20 curve within the OxCal online environment (Reimer et al., 2020).

### 3.2 Luminescence-based bleaching proxies

The following three luminescence-based bleaching proxies were used to study the sediment dynamics:

1.    Overdispersion is a measure of the spread within an equivalent dose distribution that cannot be explained by the individual uncertainties of the single-grain equivalent dose estimates such as measurement errors, parameter fitting errors and/ or measurement equipment errors. Overdispersion is often associated with incomplete bleaching due to processes like turbulent or subaquatic sediment transport under turbid conditions (Brill et al., 2018; Cunningham et

al., 2015). It is indicative for heterogeneous luminescence signal resetting and may therefore serve as a valuable bleaching proxy (Olley et al., 2004).

2.    The percentage of saturated grains can function as a proxy for the percentage of unbleached bedrock grains (Reimann et al., 2017; Bonnet et al., 2019; Guyez et al., 2022); these grains were eroded from the parent material and travelled





without sufficient exposure to reset their luminescence signal. The 2D0 criterion (85% full saturation) is applied to differentiate if a grain is in saturation or not (Wintle & Murray, 2006).

3. The Minimum Age Model (MAM) is a statistical model used to identify the palaeodose based on the population of best bleached grains within a heterogeneously bleached sample (Galbraith et al. 1999). It is typically applied to samples from fluvial settings, since fluvial samples are often heterogeneously bleached (Cunningham & Wallinga, 2012; Rhodes, 2011; Rodnight et al., 2006). A bootstrapped version of the MAM (bMAM) measures the average statistical properties of a subsample, as the relation between a subsample and a sample is considered to be the same as the relation between a subpopulation and a sample (Cunningham & Wallinga, 2012). The percentage of grains used to calculate the bMAM-equivalent dose can be identified by scanning how many grains of the total population correspond with the bMAM based palaeodose value within $2\sigma$ of their uncertainty boundaries. It is expected that the outcome of this proxy can be translated into bleaching dynamics: the lower the percentage of grains used to identify the bMAM-$D_e$, the higher the spread in the distribution.

## 4 Results

### 4.1 Measurement protocol performance

Test measurements of the extracted quartz-rich fraction on two samples (Pk-B04 and Pk-E16) showed that the quartz grains contained in the Pokhara gravels are not suitable for OSL dating, mainly because of a poor luminescence sensitivity and the absence of the fast OSL component; two out of 540 grains (<1%) held a measurable luminescence signal. The fast ratios of 0.94 and 3.38 of these two grains, calculated according to Durcan and Duller (2011), are significantly lower than the suggested threshold of 20. Thus, all subsequent measurements were performed on feldspar minerals, of which the IRSL-50 and pIRIR-150 signals showed clearly defined decay curves (Fig. 3a and b). The residual dose corrected dose recovery ratio was calculated (App. A) as $0.88 \pm 0.01$ (n = 68 grains) and $0.95 \pm 0.01$ (n = 51 grains) for the IRSL-50 and pIRIR-150 signal, respectively. The fading g-value was measured at $7.20 \pm 0.89$ %/decade and $0.89 \pm 0.87$ %/decade for the IRSL-50 and pIRIR-150 signal, respectively.

### 4.2 Single-grain feldspar burial ages

Table 2 shows the estimated paleodose (based on bMAM-$D_e$), the dose rate, the (non-)faded burial ages, and the overestimation ratio of the IRSL-50 and pIRIR150 age for each sample. The palaeodose derived from the bMAM dose estimate ranges from $4.95 \pm 0.64$ to $203 \pm 14.2$ Gy and from $34.0 \pm 6.32$ to $337 \pm 53.8$ Gy for the IRSL-50 and the pIRIR150, respectively. The dose rates range from $2.46 \pm 0.02$ to $3.74 \pm 0.04$ Gy ka$^{-1}$ with no clear pattern in function of run-out distance. However, the dose rates of samples taken from the Phusre and Saraudi tributary valleys (Pk-X17, Pk-B04 and Pk-D15) are ~30% lower than dose rates from the Seti main valley, presumably reflecting on the different mineralogy of the sourced sediments. The IRSL-50 signal consistently results in lower burial ages than the pIRIR-150 data, and therefore in a lower overestimation ratio of the



radiocarbon benchmark. Fading corrected burial ages range from $4.70 \pm 1.23$ to $205 \pm 88.9$ and from $10.0 \pm 2.19$ to $136 \pm 15.9$
ka for the IRSL-50 and the pIRIR150, respectively.

### 4.3 Overestimation of the geochronological benchmark

To further support the radiocarbon dating benchmark we took two radiocarbon samples from the Pokhara Formation (Table 3)
in addition to the radiocarbon dataset presented by Schwanghart et al. (2016). The obtained age was calibrated to $736 \pm 52$ and
$823 \pm 87$ years BP for sample Pk_DC-1401 and Pk_DC-1402, respectively. Two luminescence samples, Pk-D13 and Pk-D14,
were sampled in close proximity to the radiocarbon samples Pk_DC_1401 and Pk_DC_1402 (Fig. 4). The fading corrected
burial age of sample Pk-D13 is $13.2 \pm 5.62$ and $42.8 \pm 22.4$ ka for the IRSL-50 and the pIRIR-150, respectively. The fading
corrected burial age of sample Pk-D14 is $41.1 \pm 11.7$ and $33.3 \pm 9.84$ ka for the IRSL-50 and the pIRIR-150, respectively.

All luminescence burial ages overestimate the median value of the radiocarbon benchmark (Fig. 5), with an average
ratio of ~23 and ~72 for the IRSL-50 and pIRIR-150, respectively. The fading corrected burial ages, for both signals, result in
even higher overestimation ratios. The smallest overestimation ratios occur for the IRSL-50 burial ages of samples PK-A02
and PK-E16 with ~2.65 and ~1.51, respectively. No clear relationship between overestimation ratios and the distance between
sample location and fan apex can be observed (Fig. 5). The larger difference between fading corrected and uncorrected burial
ages for IRSL-50 compared to the pIRIR-150 signal is a result of the larger IRSL-50 fading rates. The consistent differences
between burial ages and resulting overestimation ratios of the IRSL-50 and pIRIR-150 ages compared to the radiocarbon
benchmark suggest that our samples are strongly affected by incomplete bleaching (Kars et al., 2014). The large scatter in the
radial plots shown in figure 6 likewise suggest that the IRSL-50 signal of our samples is significantly impacted by
heterogeneous bleaching. Populations of grains can be homogeneously or heterogeneously bleached. Heterogeneously
bleached populations contain zeroed grains as well as grains with remnant doses (Duller., 2008).


### 4.4 Using proxies to identify sediment dynamics

Table 4 lists the overdispersion, the number of accepted and saturated grains, and the calculated bleaching proxies of
saturated, best-bleached, and other grains for all samples and both feldspar signals (for details see section 3.2). The data of the
three bleaching proxies always sum up to 100 percent. Figure 7a shows the relationship between overdispersion and sample
location distance downstream from the apex. A trend towards higher overdispersion values is observed for the IRSL-50 signal,
except for sample Pk-B03, which suggests that IRSL-50 is more heterogeneously bleached than the pIRIR-150 signal. Fig. 7b
shows the relationship between overdispersion and equivalent dose. For both signals, overdispersion decreases with increasing
equivalent dose values. This suggests that heterogeneous bleaching is the main source of scatter in the corresponding equivalent
dose distributions (Fig. 6a-d) for both feldspar signals.

The stack graphs in Fig. 8 show the percentage of grains attributed to each bleaching proxy over the fan length for
the IRSL-50 (Fig. 8a) and pIRIR-150 (Fig. 8b) signal, respectively. The percentage of best-bleached grains is comparable for





both signals, although the IRSL-50 signal shows slightly higher percentages. The percentage of saturated grains for the pIRIR-150 signal is constantly higher than for the IRSL-50 signal, which is an artefact of the high IRSL-50 fading rates - i.e., an athermal signal loss during burial. The high numbers of pIRIR-150 saturated grains suggest a high input of bedrock grains. At 25 kilometres from the apex at the Annapurna massif a dip in percentage of saturated grains for both signals are visible. For the IRSL-50 data the reduction in percentage of saturated grains is from ~ 70 to 37 whereas the for the pIRIR150 data the reduction is only from ~ 97 to 88.

## 5 Discussion

### 5.1 Dating mass movements using feldspar single-grains

As expected from previous studies (e.g., Jaiswal et al., 2009), the investigated quartz minerals contained in the Pokhara Formation are largely unsuitable for single-grain luminescence dating due to their poor luminescence sensitivity. Less than 1% of the grains yielded a sensitive quartz OSL signal which was not dominated by the fast OSL signal component. Feldspar minerals are more sensitive and were therefore chosen as the main study material. Single-grain feldspar burial ages show that the IRSL-50 consistently results in lower burial ages than the pIRIR-150 signal. It should be considered, however, that the IRSL-50 signal suffers from a relatively high fading rate of 7.20 ± 0.89 %/decade whilst the pIRIR-150 signal is subject to a lower fading rate of 0.89 ± 0.87 %/decade, and therefore the gap between the fading corrected burial ages becomes relatively smaller. The IRSL-50 signal of two out of ten samples yielded a Late-Holocene age of the Pokhara Formation, yet still overestimating the ages obtained from an extensive collection of radiocarbon samples (Schwanghart et al., 2016, Stolle et al., 2017) by 2.65 and 1.51, respectively. The IRSL-50 signal of an additional three samples dated the fan at Early Holocene age; and the other samples indicated a Pleistocene age. The pIRIR-150 signals greatly overestimate the radiocarbon benchmark and none of the samples corresponds with the expected Holocene mass movement depositional age within assigned 2-sigma uncertainties.

Our dating results generally confirm previous luminescence studies with samples taken from high mountainous settings, i.e., fast sediment production and transfer (e.g., Scherler et al., 2015; Bonnet et al., 2019). The chance for full daylight exposure under these conditions is very limited, resulting in a vast majority of feldspar grains being insufficiently reset by daylight prior to burial. Yet, unlike for samples from the Rangitikei River in New Zealand analysed in Bonnet et al. (2019) the amount of well-bleached grains from the Pokhara formation was too limited to be able to establish a robust absolute chronology based on feldspar single-grain luminescence analyses. In the case of the long run-out mass movement from Pokhara only ~2% (Pk-A02) and ~8% (Pk-E16) of the grains from the two best-bleached samples [n = 10] yield ages within the targeted age range. It should be noted, however, that the sampled sediments from the Pokhara Formation presumably represent one of the most extreme settings in terms of bleaching likelihood during sediment transport as several km$^3$ of sediment were mobilized in the headwaters of the upper Seti River (Schwanghart et al., 2016, Lavé et al., 2022). In other, less extreme, high mountainous settings the application of feldspar single-grain to Late Glacial or Holocene samples proved to be successful for most samples



for both the IRSL-50 (e.g., Scherler et al., 2015) and the low-temperature pIRIR (e.g., van Gorp et al., 2013; Bonnet et al., 2019).

**5.2 Age range finder**

Our observation that the IRSL-50 signal of two out of ten samples captured the Late Holocene age while none of the samples shows pIRIR-150 ages in line with the Late Holocene further highlights the expected extremely fast mobilization and high-turbidity transport mechanisms under which the sediments from the Pokhara Formation were formed. It is well established that the IRSL-50 signal bleaches quicker than the pIRIR-150 signal (Kars et al., 2014) which suggests that the pre-exposure (prior to the extreme event) and transport time was too limited and/or the turbidity of the discharge was too high (Mey et al., 2023) and/or the deposition rate was too high to provide sufficient light exposure to fully bleach the IRSL-50 signal and at least partly bleach the pIRIR-150 signal of a significant number of grains (Brill et al., 2018). Lavé et al. (2022) sampled material from the internal shear zone high up in the Sabche Cirque (Fig. 1a) and assumed that increased temperature and/or shear heating sufficiently reset the IRSL-50 and pIRIR-225 signals for dating, a phenomenon referred to as triboluminescence. Like our data, their IRSL-50 age distributions also show highly scattered data with a minimum age of ~900 years. Their scattered age distributions also suggest that the samples are heterogeneously reset, suggesting that shear heating was not able to fully reset the OSL signal (Bateman et al., 2018). Regardless, the recent results of Lavé et al. (2022) and our results indicate that luminescence methods have the potential to be used as an age range finder in dating these extremely dynamic mass movement events. If the main research goal is to place similar mass movements into a general stratigraphic context, for example to establish whether the valley infill is of Holocene or Pleistocene (early, mid or late) age, feldspar single-grain IRSL-50 will facilitate a correct chronostratigraphic interpretation. However, the data must be interpreted cautiously, and care must be taken to retrieve the youngest subpopulation from a sample by applying the bMAM. Yet even the youngest subpopulation may still overestimate the true age by a few thousand years. In our case the IRSL-50 signal of 2 out of 10 samples yielded the correct Late Holocene age range. Therefore, we advise to study at least 10 samples when dating samples from comparable mass movement deposits. It should also be noted that the advantages of using single-grain feldspar IRSL-50 as an age range finder method, compared to conventional quartz OSL dating, are (i) the availability of a sensitive luminescence signal even in settings in close proximity to plutonic or metamorphic bedrock (e.g. Guralnik et al., 2015) and (ii) the very measurement-time-efficient data gathering, as typically ~50 % of the K-rich feldspar signal provides a suitable IRSL/pIRIR luminescence response (e.g. Reimann et al., 2017).

**5.3 Using luminescence-based proxies for tracing sediment dynamics**

The heterogeneous nature of the sediments is manifested in high overdispersion values ranging from 43 to 120 % and 14 to 88 % for the IRSL-50 and pIRIR-150 signal, respectively. The lower overdispersion of the pIRIR-150 signal should thereby be interpreted as a result of more homogeneous poor bleaching compared to the on average more heterogeneous poor bleaching of the IRSL-50 signal. Additionally, the relationship between overdispersion and equivalent doses (Fig. 7b) shows, for both





signals, a decrease of overdispersion with increasing equivalent dose values, which suggests bleaching as the main source of scatter in the corresponding equivalent dose distributions (Brill et al., 2018; Cunningham et al., 2015). Interestingly, we could not observe a trend of the overdispersion with run-out distance, i.e., the degree of heterogeneous bleaching neither increases nor decreases with distance from the apex. Again, this hints at extremely limited bleaching opportunities overall, and thus

mobilisation of sediments with very limited pre-exposure, very rapid sediment transport under high turbidity conditions and immediate burial after deposition (i.e., very high sedimentation rates).

Apart from a dip in in the percentage of saturated grains at ~25 km distance from the apex (Fig. 8a), especially in the IRSL-50 data, the results of the other two bleaching proxies show no clear trend with run-out distance. Guyez et al. (2022) propose that a decrease of the fraction of saturated grains may either indicate a lower input of bedrock grains or a change in

fluvial bleaching potential. Bearing in mind the catastrophic nature of sediment transport that resulted in the emplacement of the Pokhara formation, a change of the fluvial bleaching potential seems unlikely, and we interpret that the dip in the percentage of saturated grains more likely indicates a change in the sediment contribution of different source areas. While saturated grains are typically sourced from side gorges and/or local valley wall (bedrock) incisions, the decrease of the percentage of saturated grains may indicate the contribution of fluvially transported sediments from a tributary of the Seti River. In our case the most

likely explanation is the activation of previously deposited fluvial sediments from the Bijayapur Khola (Khola means small river in Nepali) (Fig. 1b).

Our data of the three bleaching proxies (Fig. 8a and b) lack an overall trend with runout distance. Absence of clear spatial relationships suggests that sediments were transported under conditions with an extremely low probability of bleaching which points at fast mobilization of the grains, transport under highly turbid conditions and low chances for sediment cycling in the

floodplain. This strongly supports the hypothesis put forward by Stolle et al. (2017) that Pokhara sediment fill-in was deposited by one or several discrete mass movement events.

McGuire and Rhodes (2015a), Gray (2017) and Guyez et al. (2023) used single-grain luminescence data to calculate virtual velocities for grains transported in fluvial systems; a virtual velocity is a velocity averaged over transport and rest times (Gray et al., 2019). The estimated virtual velocities are based on bleaching during in-channel transport. These studies show

along-stream decrease of the luminescence signals for their less extreme mountainous settings; additionally, McGuire and Rhodes (2015b) showed that differential luminescence decreases depending on IR/pIRIR stimulation temperature. Due to the absence of a clear trend of luminescence proxies with run-out distance, our single-grain luminescence data cannot be used to calculate the virtual velocity of the associated sediments. We encourage future work on river sediment systems that can be placed between the dynamic mountainous settings studied by McGuire and Rhodes (2015) and Guyez et al. (2022, 2023) and

our extreme mass movement. This would help to identify the potential threshold between clear longitudinal trends of luminescence proxies for dynamic mountainous settings and the absence of a clear trend for extreme mass movement sediments. We propose to mainly focus on the proxies best-bleached and saturated grains in future work such as Guyez et al. (2023), and not on the overdispersion proxy. Our data and previous studies showed (Guyez et al., 2023) that the overdispersion





proxy is less sensitive in capturing bleaching dynamics and longitudinal trends then the fraction of saturated and best-bleached
grains.

## 6 Conclusion

This study tested the application of single-grain (feldspar) luminescence for dating and reconstructing sediment dynamics of
an extreme mass movement event in the Himalayan Mountain range. Our analysis revealed that the quartz signal did not bear
a sensitive signal, which strongly limits the suitability of the method for determining the age of Himalayan mass movement
deposits. The IRSL-50 and pIRIR-150 feldspar signals were used for estimating the age range of the deposits and underscored
the expected extreme fast mobilization and high-turbidity transport mechanisms under which the Pokhara Formation was
emplaced. If one's main research goal is to place similar mass movements into a general stratigraphic context, feldspar single-
grain IRSL-50 will lead to the most plausible chronostratigraphic interpretation. Care must be taken to focus on retrieving the
youngest subpopulation from a sample by applying the bMAM. Age overestimated by a few thousand years can, however, still
occur.

The high overdispersion values underline the heterogeneous nature of the sediment and the decrease of overdispersion
with increasing equivalent dose implies that the degree of bleaching is the main source of scatter in the equivalent dose
distributions for both feldspar signals under study. Our data shows that overdispersion is less sensitive in capturing longitudinal
trends than our two bleaching proxies; therefore, we propose to focus on studying proxies based on best-bleached and saturated
grains in future work. These two bleaching proxies show no general longitudinal trend either. However, they capture a
decreased input of saturated bedrock grains at ~25 kilometres from the apex, which are possibly related to a change in sediment
contribution from different source areas.

The absence of clear spatial relationships suggests that sediments were transported under conditions with extremely
low bleaching probabilities, which prevented us from calculating virtual velocities. Still, our data support rather than falsify
the current hypothesis that the Pokhara sediment fill-in was deposited by one or several extreme mass movement events.

## Appendices

**Appendix A: Overview of residual and recovered doses for both signals, and the resulting dose recovery ratio. The fading g-value is also stated for both signals, as well as the number of grains contributing to the calculations.**

| IRSL-50 | Residual dose | Dose recovery | pIRIR-150 | Residual dose | Dose recovery |
|---|---|---|---|---|---|
| CAM-De [Gy] | 0.43 ± 0.06 (n=29 grains) | 13.7 ± 0.20 (n=68 grains) | CAM-De [Gy] | 1.18 ± 0.15 (n=12 grains) | 15.4 ± 0.30 (n=51 grains) |
| Dose recovery ratio | 0.88 ± 0.01 | | Dose recovery ratio | 0.95 ± 0.01 | |



| Fading g-value | 7.20 ± 0.89%/decade | Fading g-value | 0.89 ± 0.87 %/decade |
|---|---|---|---|

**Code and data availability**

The code and data to run the analysis are available in the supplemental data.

**Author contributions**

All authors contributed to the study conception and design. Fieldwork was performed by WS, JM, BRA, AMB. Laboratory work, measurements and data analyses were performed at the Netherlands Centre for Luminescence dating at Wageningen University & Research by AMB with the support of TR. The first draft of the manuscript was written by AMB, and all authors

commented on previous versions of the manuscript. All authors read and approved the final manuscript.

**Competing interests**

The authors declare that they have no conflict of interest.

**Acknowledgements**

The authors gratefully acknowledge funding from the VolkswagenStiftung for the Experiment!-project "Illuminating the speed

of sand - quantifying sediment transport using optically stimulated luminescence". The samples Pk-X17 and 18 were collected by Amelie Stolle (University of Osnabrück), John Jansen (Czech Academy of Sciences), Oliver Korup (University of Potsdam) and Tim J. Cohen (University of Wollongong), whom we thank for their contribution to this work. We thank Alice Versendaal and Erna van den Hengel-Voskuilen for their support at the Netherlands Centre for Luminescence dating.

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



**Figures**



Figure 1: Map of the Pokhara Valley in Central-Nepal (a), displays the sampled locations (white round symbols), and the Pokhara and Ghachok Formation. Map in panel (b) shows the distance from the fan apex in the Pokhara valley, and all river tributaries and their confluences (black square symbols). Pictures (c-f) illustrate sample locations Pk-E16 (c), Pk-C05 (d), Pk-B03 (e), and Pk-D15 (f).








**Figure 2: Photographical impression of the incised Pokhara Formation (a-c), the Seti River, and its tributaries. The snow-covered mountain range depicts the Annapurna Massif; the highest mountain peak (c) is the Machapuchare locally known as the fishtail mountain.**

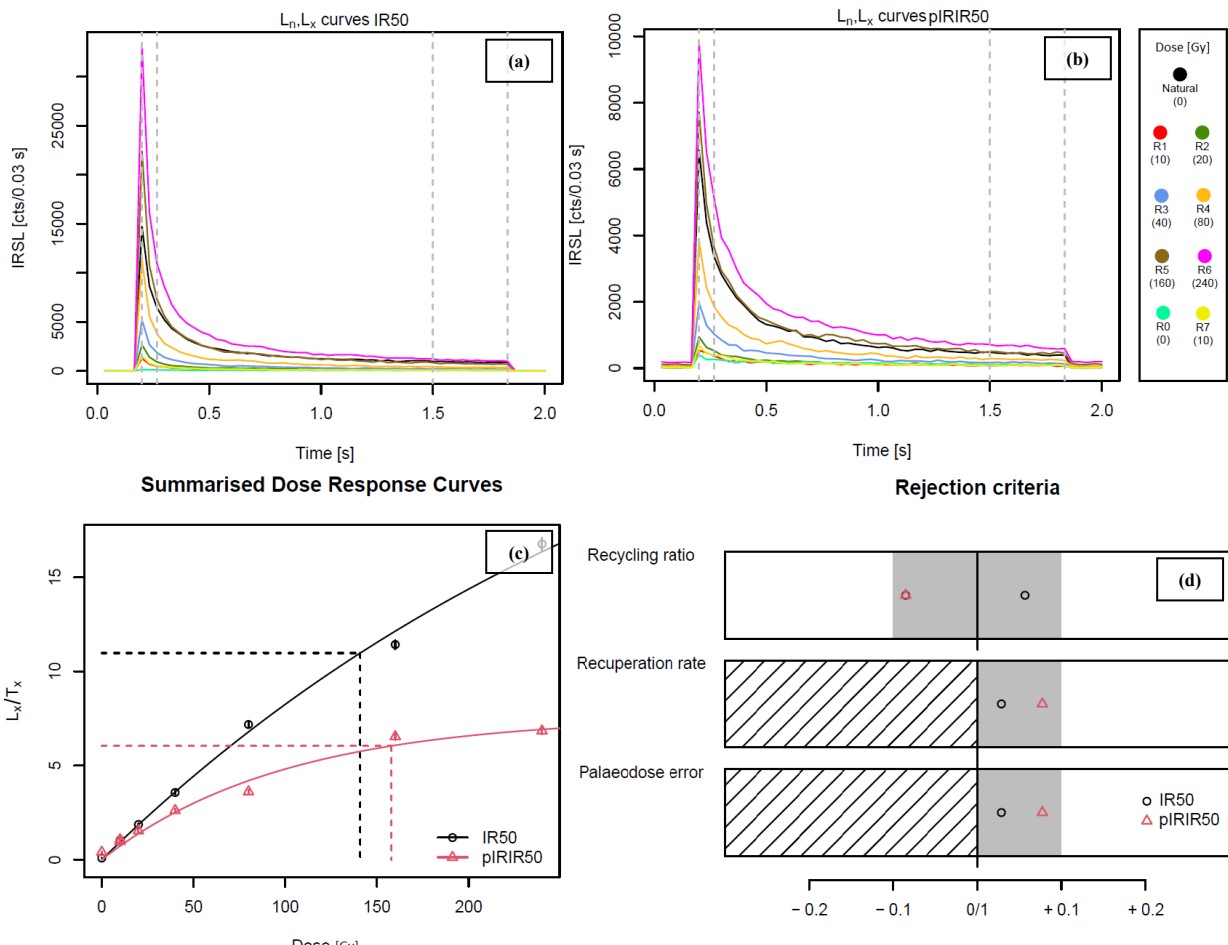

**Figure 3: Panel a and b show the decay curves for measured IRSL-50 and pIRIR-150 signals, respectively. The black line shows the natural decay curves, whilst the coloured lines show the decay curves in response to the regenerative doses as indicated in the legend. Panel c shows the summarized dose response curves for both signals; the dotted line indicates the estimated equivalent dose on the x-axis in Gray. Panel d shows that both signals are within unity (± 10%) for the three rejection criteria: recycling ratio, recuperation rate, and paleodose error.**




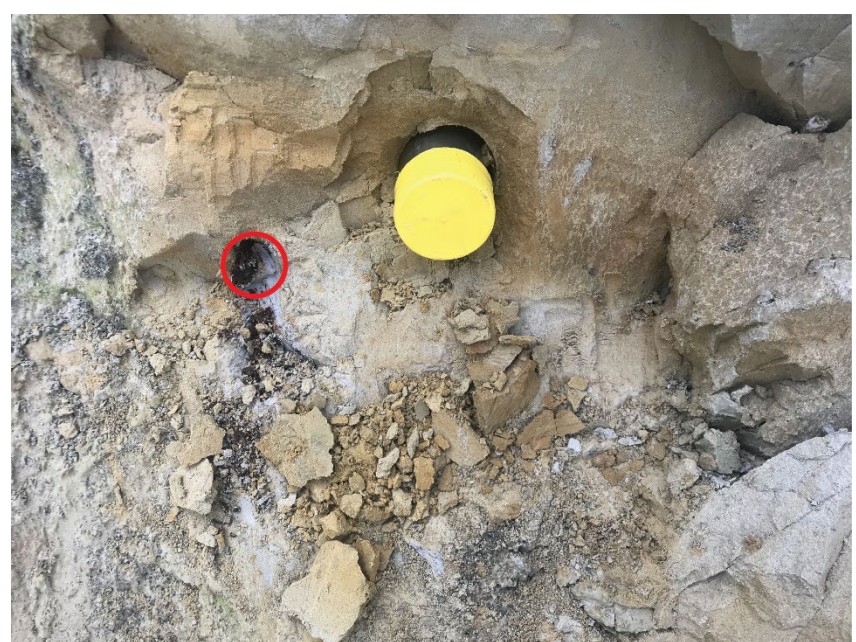

**Figure 4: A field photograph of the outcrop where the radiocarbon sample Pk_DC_1402 was taken as circumcised by the red circle; the yellow-capped OSL sample tube with a diameter of five centimetre corresponds to sample Pk-D13. For location see Fig. 1a.**

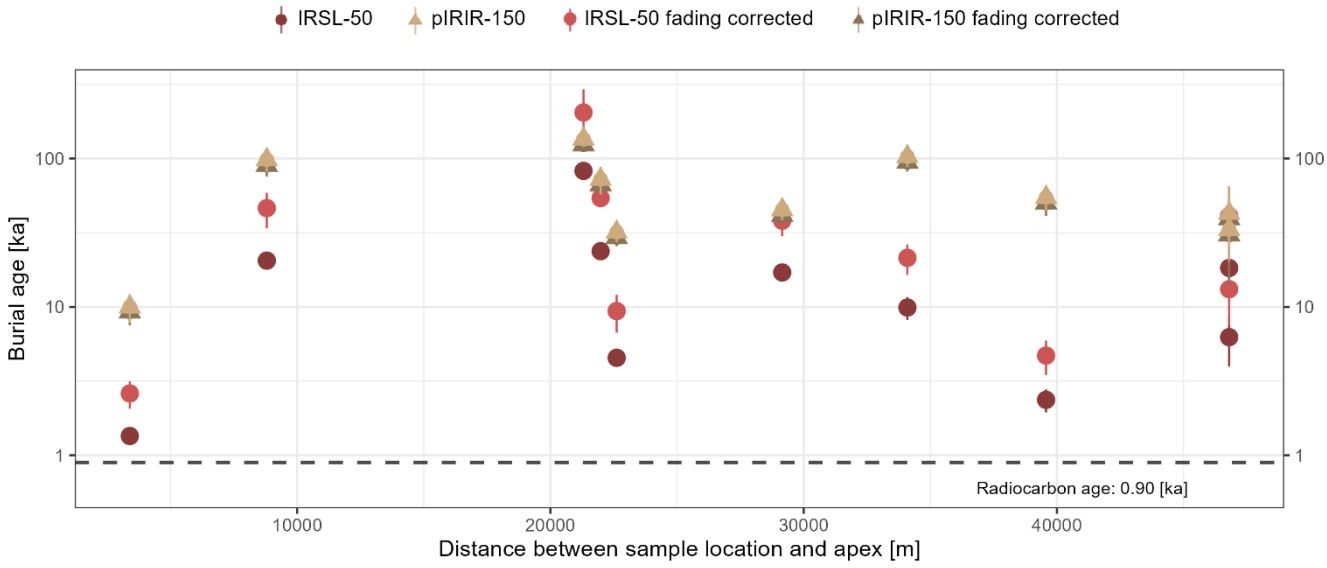


**Figure 5: The obtained burial ages and their associated uncertainties (1-sigma confidence interval) are plotted against the distance between sample location and the fan apex of the mass movement [m]. The dark red dots correspond to the IRSL-50 signal, the light red dots to the fading corrected IRSL-50 signal values. The light brown triangles correspond to the pIRIR-150 signal, the dark brown triangles to the fading corrected pIRIR-150 signal values. The dashed horizontal line indicates the benchmark age of 0.90 ka**

**established based on an elaborate radiocarbon chronology. Note the y-axis is plotted on logarithmic scale.**





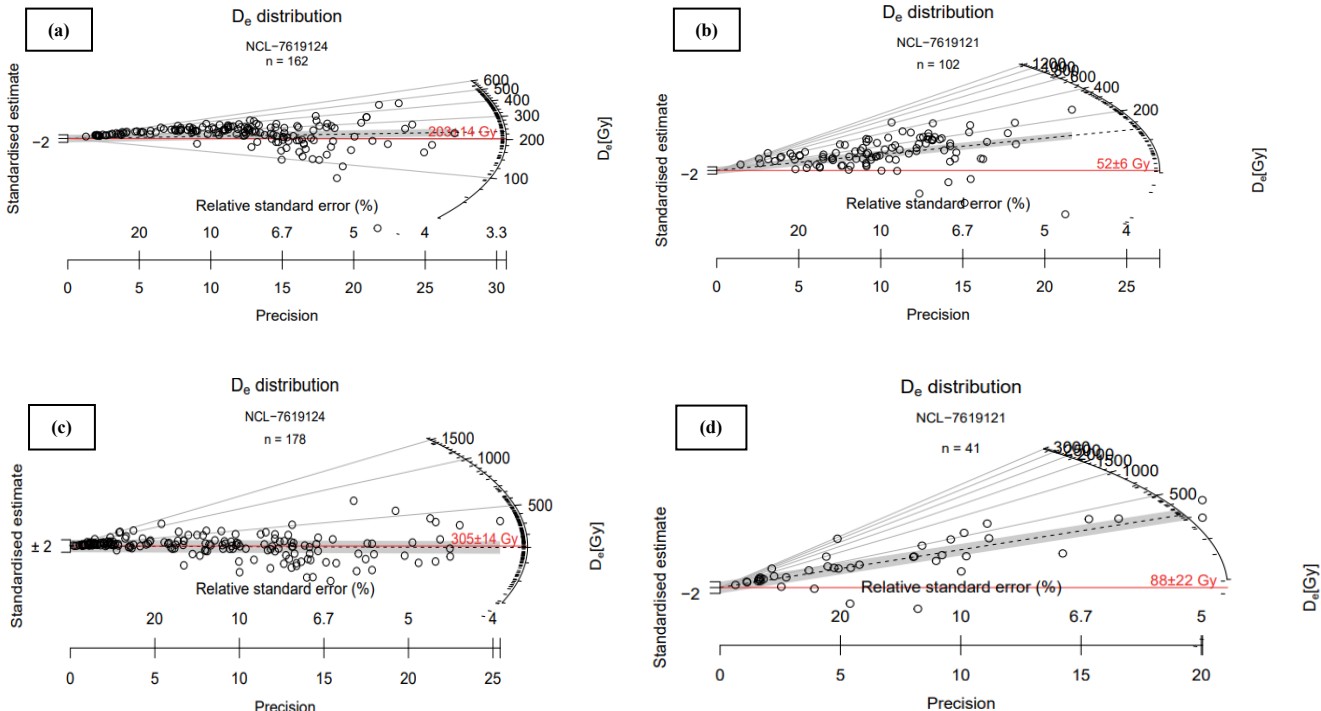

**Figure 6: The radial plots in panel a and b show the equivalent dose distributions of the IRSL-50 and pIRIR-150 signal respectively for sample Pk-X17 (NCL-7619124) and the radial plots in panel c and d show the same for sample Pk-X18 (NCL-7619121). The red line indicates the palaeodose in Gray as estimated by a bootstrapped version of the Minimum Age Model (Cunningham and Wallinga, 2012). n indicates the number of grains per radial plot.**

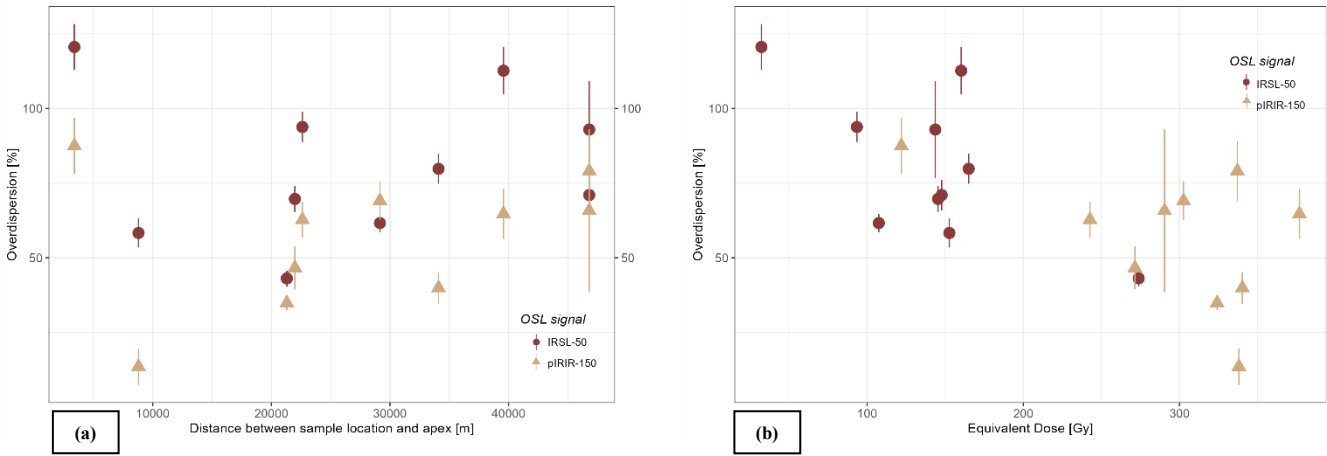

**Figure 7: Panel a shows the relationship between overdispersion [%] and the distance between the sample location and the fan apex of the mass movement expressed in meters. Panel b shows the relationship between overdispersion [%] and equivalent dose [Gy]. The dark red dots correspond to the IRSL-50 signal, the light brown triangles to the pIRIR-150 signal.**




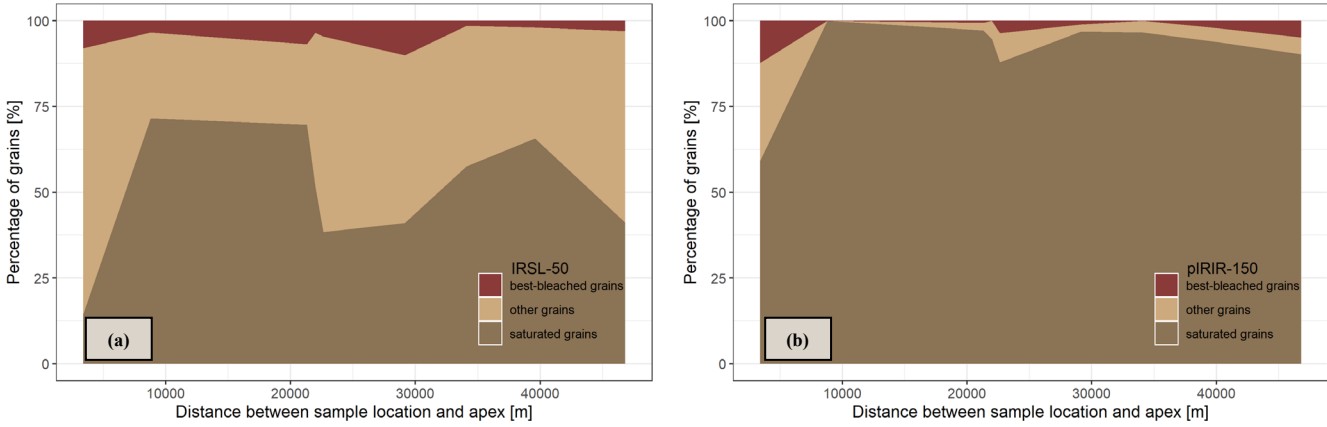


**Figure 8: Stack graphs showing the relationship between the percentage of grains for each of our three bleaching proxies, e.g. best-bleached grains in dark red, saturated grains in dark brown, and other grains in light brown, and the distance between the sample location and the fan apex of the mass movement. Panel a shows the IRSL-50 data, panel b displays the pIRIR-150 data. Other grains**

**represent grains that are incompletely bleached and/ or have an inherited dose.**



**Tables**

Table 1: Measurement protocols applied within this study: details on single-grain equivalent dose and dose recovery measurements on the left side, and details on the multi-grain fading test on the right side of the table. Lx indicates the natural or regenerative OSL dose obtained and is corrected by the OSL response to the test dose Tx.

| Treatment single-grain De and dose recovery tests | Treatment multi-grain fading test | Measurement |
|---|---|---|
| Natural or regenerative dose | Natural or regenerative dose | |
| Preheat at 175˚ for 120 s | Preheat at 175˚ for 120 s | |
| SG infrared stimulation at 50˚ for 2 s | MG infrared stimulation at 50˚ for 100 s | Lx – IRSL-50 |
| SG infrared stimulation at 150˚ for 2 s | MG infrared stimulation at 150˚ for 100 s | Lx – pIRIR-150 |
| Test dose: 10 Gy | Test dose: 5 Gy | |
| Preheat at 175˚ for 120 s | Preheat at 175˚ for 120 s | |
| SG infrared stimulation at 50˚ for 2 s | MG infrared stimulation at 50˚ for 100 s | Tx – IRSL-50 |
| SG infrared stimulation at 150˚ for 2 s | MG infrared stimulation at 150˚ for 100 s | Tx – pIRIR-150 |
| IR wash at 150˚ for 500 s | MG infrared stimulation at 150˚ for 500 s | |
| Repeat all above steps for irradiation [Gy]: Natural – 10 – 20 – 40 – 80 – 160 – 240 – 0 – 10 [De] 15 – 10 -20 – 40 – 80 – 0 – 25 [Dose Recovery] | Repeat all above steps for irradiation [Gy]: Natural – 10 – 10 – pause 500 s – 10 – 10 – 10 | |

Table 2: A tabular overview of all single-grain feldspar equivalent doses, dose rates, (fading corrected) burial ages, overestimation ratios, and their associated uncertainties (1-sigma confidence interval) obtained within this study. All luminescence ages refer to the date of sampling: October 2019.

| Lab code | Sample ID | Coordinates [Lat / Lon] | Signal | bMAM De [Gy] | Dose rate [Gy/ka] | Burial age faded[ka] | Burial age non-faded [ka] | Overestimation ratio [-] |
|---|---|---|---|---|---|---|---|---|
| NCL-7619124 | Pk-X17 | 28.1828 83.9526 | IRSL-50 pIRIR-150 | 203 ± 14.2 305 ± 14.2 | 2.46 ± 0.02 | 82.6 ± 5.83 124 ± 5.87 | 205 ± 89.0 136 ± 15.9 | 92.5 138 |
| NCL-7619122 | Pk-D15 | 28.02022 84.04914 | IRSL-50 pIRIR-150 | 29.5 ± 5.20 281 ± 37.9 | 2.97 ± 0.03 | 9.91 ± 1.75 94.5 ± 12.8 | 21.4 ± 4.98 103.2 ± 15.7 | 11.1 106 |
| NCL-7619109 | Pk-A02 | 28.01487 84.09628 | IRSL-50 pIRIR-150 | 8.28 ± 1.47 176 ± 32.5 | 3.50 ± 0.03 | 2.36 ± 0.42 50.3 ± 9.28 | 4.70 ± 1.23 54.7 ± 10.0 | 2.65 56.3 |
| NCL-7619123 | Pk-E16 | 28.34406 83.96840 | IRSL-50 pIRIR-150 | 4.95 ± 0.64 34.0 ± 6.32 | 3.67 ± 0.03 | 1.35 ± 0.18 9.26 ± 1.72 | 2.61 ± 0.55 10.0 ± 2.19 | 1.51 10.4 |
| NCL-7619111 | Pk-B04 | 28.17621 83.97538 | IRSL-50 pIRIR-150 | 64.6 ± NA 181 ± 30.7 | 2.72 ± 0.03 | 23.8 ± NA 66.6 ± 11.3 | 54.2 ± NA 72.6 ± 15.3 | 26.6 74.6 |
| NCL-7619125 | Pk-X18 | 28.17000 83.99800 | IRSL-50 pIRIR-150 | 16.6 ± 1.73 107 ± 13.5 | 3.65 ± 0.03 | 4.54 ± 0.48 29.4 ± 3.70 | 9.38 ± 2.68 31.9 ± 4.70 | 5.08 32.9 |
| NCL-7619112 | Pk-C05 | 28.30042 83.92705 | IRSL-50 pIRIR-150 | 76.8 ± 9.77 337 ± 53.8 | 3.74 ± 0.04 | 20.5 ± 2.62 90.2 ± 14.4 | 46.4 ± 12.5 98.4 ± 17.3 | 23.0 101 |
| NCL- | Pk-B03 | 28.10841 | IRSL-50 | 52.0 ± 4.10 | 3.05 ± 0.02 | 17.1 ± 1.35 | 38.1 ± 8.20 | 19.1 |



| 7619110 | | 84.11581 | pIRIR-150 | 127 ± 16.7 | | 41.5 ± 5.47 | 45.2 ± 6.88 | 46.5 |
| NCL- | Pk-D14 | 27.96179 | IRSL-50 | 52.4 ± 6.04 | 2.86 ± 0.03 | 18.3 ± 2.12 | 41.1 ± 11.7 | 20.5 |
| 7619121 | | 84.11234 | pIRIR-150 | 87.6 ± 22.4 | | 30.6 ± 7.82 | 33.3 ± 9.84 | 34.3 |
| NCL- | Pk-D13 | 27.96179 | IRSL-50 | 19.1 ± 6.97 | 3.05 ± 0.03 | 6.26 ± 2.28 | 13.2 ± 5.62 | 7.01 |
| 7619120 | | 84.11234 | pIRIR-150 | 120 ± 78.3 | | 39.3 ± 25.6 | 42.8 ± 22.4 | 44.0 |

**Table 3: A tabular overview of the newly dated radiocarbon samples, including the raw age and the calibrated age.**

| Sample ID | Lab ID | Coordinates [Lat/ Lon] | Age [y BP] | Age_error [y BP] | Cal_age [cal y BP] | Cal_age_error [cal y BP] |
|---|---|---|---|---|---|---|
| Pk_DC_1401 | Poz-120378 | 27.96200 84.11200 | 840 | 30 | 736 | 52 |
| Pk_DC_1402 | Poz-120379 | 27.96200 84.11200 | 910 | 30 | 823 | 87 |

**Table 4: A tabular overview of all data associated with the calculated bleaching proxies expressed in numbers (#) or percentage (%).**

| Lab code | Signal | #accepted grains | #saturated grains | %grains in saturation | %grains best-bleached | %other grains | Overdispersion [%] | Distance from apex [m] |
|---|---|---|---|---|---|---|---|---|
| NCL-7619124 | IRSL-50 | 162 | 113 | 69.8 | 6.79 | 23.5 | 43.1 ± 2.63 | 21308 |
| | pIRIR-150 | 178 | 173 | 97.2 | 0.56 | 2.25 | 34.9 ± 2.44 | |
| NCL-7619122 | IRSL-50 | 139 | 80 | 57.6 | 1.44 | 41.0 | 79.8 ± 5.02 | 34098 |
| | pIRIR-150 | 59 | 57 | 96.6 | 0.0 | 3.39 | 39.9 ± 5.27 | |
| NCL-7619109 | IRSL-50 | 105 | 69 | 65.7 | 1.90 | 32.4 | 113 ± 8.00 | 39574 |
| | pIRIR-150 | 51 | 48 | 94.1 | 1.96 | 3.92 | 64.7 ± 8.42 | |
| NCL-7619123 | IRSL-50 | 125 | 18 | 14.4 | 8.0 | 77.6 | 121 ± 7.70 | 3393 |
| | pIRIR-150 | 49 | 29 | 59.2 | 12.2 | 28.6 | 87.5 ± 9.40 | |
| NCL-7619111 | IRSL-50 | 143 | 74 | 51.8 | 3.50 | 44.8 | 69.7 ± 4.34 | 21984 |
| | pIRIR-150 | 38 | 36 | 94.7 | 0.0 | 5.26 | 46.6 ± 7.14 | |
| NCL-7619125 | IRSL-50 | 177 | 68 | 38.4 | 4.52 | 57.1 | 93.8 ± 5.07 | 22618 |
| | pIRIR-150 | 83 | 73 | 88.0 | 3.61 | 8.43 | 62.8 ± 5.96 | |
| NCL-7619112 | IRSL-50 | 88 | 63 | 71.6 | 3.41 | 25.0 | 58.3 ± 4.87 | 8802 |
| | pIRIR-150 | 10 | 10 | 100 | 0.0 | 0.0 | 13.6 ± 6.11 | |
| NCL-7619110 | IRSL-50 | 219 | 90 | 41.1 | 10.1 | 48.9 | 61.7 ± 3.03 | 29154 |
| | pIRIR-150 | 98 | 95 | 96.9 | 1.02 | 2.04 | 69.1 ± 6.38 | |
| NCL-7619121 | IRSL-50 | 102 | 42 | 41.2 | 2.94 | 55.9 | 71.0 ± 5.16 | 46804 |
| | pIRIR-150 | 41 | 37 | 90.2 | 4.88 | 4.89 | 79.1 ± 10.1 | |
| NCL-7619120 | IRSL-50 | 17 | 10 | 58.8 | 5.88 | 35.3 | 93.0 ± 16.2 | 46804 |
| | pIRIR-150 | 4 | 3 | 75.0 | 25.0 | 0.0 | 65.8 ± 27.3 | |
