# Peer review of "Insight into the dynamics of a long run-out mass movement using single-grain feldspar luminescence in the Pokhara valley, Nepal"

_Geochronology, 2023_

## Referee Comment (RC1)

Dear editor and authors,

this study by de Boer et al. identifies a gap in knowledge and an opportunity to explore a known method and its applicability to questions that haven't been addressed before. It is a well-designed study and an overall well-written manuscript. The background and methods of this study are explained sufficiently and concisely. Results are summarised in a logical manner. The discussion and conclusions drawn are sensible and understandable. There are a few technical things that I think should be addressed.

First, the use of the phrase "OSL dating" (e.g., line 87) or "OSL signal" (e.g., line 268). The authors use this term to refer to all their measurements and signals (quartz AND feldspar). I think this might be confusing for some readers. To me, OSL dating/signal refers to the dating of or signal from quartz only. I would suggest the term "optical dating" in places where the authors mean to refer to both quartz and K-feldspar measurements. In places where only the signal from K-feldspar is being discussed, the term "IRSL signal" might be better.

Second, I think it would be helpful to expand on how the dose rate was obtained. What are the dose rate values for alpha, beta, gamma, and cosmic-ray dose rate portions? As K-feldspar samples haven't been treated with HF, was the alpha dose rate considered? A short paragraph or table in the supplementary material would be sufficient here.

The radial plots in Figure 6 need a bit of amending as for panels (b) and (d) the labels of the z-scale are very difficult to read, it almost looks like the lower De range was cut off and sometimes parts of the plots are overlain (panel d). By changing the scaling factor of the z-axis and centering the plots around the CAM estimate (instead of the bMAM), this would be improved. And the bMAM estimate could still be added as an extra line. The labels of the y-axis (standardised estimate) should all be ±2 (or +2 and -2) as shown in panel (c).

For Figure 7, I think it would be helpful to add trendlines for each of the datasets to help the reader identify the increasing or decreasing trends straight away. On first look at panel (b), it is easy to miss the "outlier" for each signal which makes it seem like there is hardly any trend visible.

The caption for Table 2 is slightly confusing - "All luminescence ages refer to the date of sampling: Oct 2019." What do you mean by that? Is there any way you can phrase this differently or explain what you mean?

And last, the authors define an acronym for organic matter "OM" (line 129). However, this acronym is defined again the exact same way in line 134 and then never used again. Since this section (3.1, lines 124 to 136) is the only place where organic matter is discussed, I think the use of the acronym is not justified and hinders the flow of the paragraph. My suggestion would be to either remove the acronym entirely or actually use the acronym (only) when it is mentioned the second time.

Overall, I think this is a promising study and I hope my comments help to improve the manuscript.

All the best,

Maria Schaarschmidt

---

## Author Response (AR1)

We thank reviewer (R1) Maria Schaarschmidt for her review, and her positive recommendation of our work.

(R1) Dear editor and authors, this study by de Boer et al. identifies a gap in knowledge and an opportunity to explore a known method and its applicability to questions that haven't been addressed before. It is a well-designed study and an overall well-written manuscript. The background and methods of this study are explained sufficiently and concisely. Results are summarised in a logical manner. The discussion and conclusions drawn are sensible and understandable. There are a few technical things that I think should be addressed.

(R1) First, the use of the phrase "OSL dating" (e.g., line 87) or "OSL signal" (e.g., line 268). The authors use this term to refer to all their measurements and signals (quartz AND feldspar). I think this might be confusing for some readers. To me, OSL dating/signal refers to the dating of or signal from quartz only. I would suggest the term "optical dating" in places where the authors mean to refer to both quartz and Kfeldspar measurements. In places where only the signal from K-feldspar is being discussed, the term "IRSL signal" might be better.

- (A) We adapted your suggestion to only use 'OSL dating' to refer to quartz measurements, however, we use the term "luminescence dating" to refer to both quartz OSL and feldspar IR-OSL and feldspar IRSL and pIRIR when only referring to feldspar luminescence measurements using IR stimulation. The term "optical dating" was introduced by the original paper of Huntley et al. (1985 in Nature) to describe optically stimulated luminescence measurements (stimulated with green at around 515 nm) mainly of quartz extracts. To use this term also for IRSL measurements on feldspars as suggested by the reviewer does not reduce the confusion in our opinion. First IRSL measurements on feldspars were reported by Hütt et al. in 1988 (in QSR) and thus 3 years after the term optical dating was introduced by Huntley.

(R1) Second, I think it would be helpful to expand on how the dose rate was obtained. What are the dose rate values for alpha, beta, gamma, and cosmic-ray dose rate portions? As K-feldspar samples haven't been treated with HF, was the alpha dose rate considered? A short paragraph or table in the supplementary material would be sufficient here.

- (A) We added a supplementary excel file in which all values for the dose rate portions are written down. Our calculations recognize an internal alpha, grain size, water attenuated, external gamma, and a cosmic contribution. In addition to that we also took external alpha and internal beta contributions into account since we are dealing with feldspar samples. Within the main text a reference to this supplementary file can be found in line 137.

(R1) The radial plots in Figure 6 need a bit of amending as for panels (b) and (d) the labels of the z-scale are very difficult to read, it almost looks like the lower De range was cut off and sometimes parts of the plots are overlain (panel d). By changing the scaling factor of the z-axis and centering the plots around the CAM estimate (instead of the bMAM), this would be improved. And the bMAM estimate could still be added as an extra line. The labels of the y-axis (standardised estimate) should all be ±2 (or +2 and -2) as shown in panel (c).

- (A) We changed the scaling factor of the z-axis for the radial plots, and made sure that all y-axis labels are correct. Please see the updated radial plots in figure 5.

(R1) For Figure 7, I think it would be helpful to add trendlines for each of the datasets to help the reader identify the increasing or decreasing trends straight away. On first look at panel (b), it is easy to miss the "outlier" for each signal which makes it seem like there is hardly any trend visible.

- (A) We added trendlines to figure 6a and figure 6b to enable a quick interpretation of the visible trends.

(R1) The caption for Table 2 is slightly confusing - "All luminescence ages refer to the date of sampling: Oct 2019." What do you mean by that? Is there any way you can phrase this differently or explain what you mean?

- (A) Luminescence ages are reported in "a" or "ka" ago and require a reference date to which this ago refers to. In other dating communities other reference dates are used such as 1950 CE (BP → 14C) , however, within the luminescence community it is common practice to refer the OSL age to the year of sampling (see e.g. Brauer et al. 2015 in QSR, section on luminescence dating).

(R1) And last, the authors define an acronym for organic matter "OM" (line 129). However, this acronym is defined again the exact same way in line 134 and then never used again. Since this section (3.1, lines 124 to 136) is the only place where organic matter is discussed, I think the use of the acronym is not justified and hinders the flow of the paragraph. My suggestion would be to either remove the acronym entirely or actually use the acronym (only) when it is mentioned the second time.

- (A) We removed the acronym for organic matter in line 129 and 134.

(R1) Overall, I think this is a promising study and I hope my comments help to improve the manuscript. All the best, Maria Schaarschmidt

We thank reviewer 2 (R2) for the thorough review of our work, the comments were very helpful in improving our manuscript. We adapted almost all 'specific comments', and clarified sentences and/ or abbreviations where necessary. You can find our response to the suggestion paragraphs below.

Suggestion 1:

(R2) The authors have dated two new radiocarbon samples in this study. However, in the section 3, material and methods, there is no information about these two radiocarbon samples. There is only one sentence at the end of this section: 'The newly obtained radiocarbon samples were calibrated with the IntCal20 curve within the OxCal online environment.' When I read this sentence, I got confused as I did not know what samples the authors are talking about. When it came to the section 4.3 in the results part, I saw the lines 'To further support the radiocarbon dating benchmark we took two radiocarbon samples from the Pokhara Formation (Table 3).' Only then did I realize that the authors have dated 2 radiocarbon samples in this study. I suggest that the authors introduce these two radiocarbon samples in the section of material and methods, and switch the orders of Figure 3 and 4.

(R2) Also, I learned from the main text that the median value of the radiocarbon dataset was used as the age benchmark (independent age control) to compare with the luminescence ages. However, I cannot find any information about this median age from the main text. Later I got this information from the figure caption of Figure 5. It is 0.90 ka. The authors should clearly state this benchmark age in the main text.

(R2) Another thing the authors should clearly state in the main text is that they used the faded ages to calculate the overestimation ratio, rather than the fading corrected ages. I got this information only from the indication in lines 199-200. I think it is worthy to add some arguments why the faded ages were chosen instead of the fading corrected ages.

- (A) Within our work we dated two newly acquired radiocarbon samples, which was indeed not very clear from the text. As you suggested we now introduce these two samples in the material and methods section. Here we also added why and how we calculated the benchmark age of the radiocarbon dataset. We moved figure 3 and added it as inset in figure 1 (g). The order of figures is now more logical to follow. In line 156 we state that we chose the faded ages to calculate the overestimation ratio. By using fading corrected ages another systematic error could be introduced.

Suggestion 2:

(R2) From the data in the supplementary excel file, I found that the authors applied the exponential + linear fitting to estimate the De and D0. I totally agree with this fitting for De estimation, as it can fit the growth curve much better than the single exp fitting. But when it comes to D0 evaluation, I feel that the single exp fitting seems to be more reasonable. When applying the exp+linear fitting, the D0 is usually much smaller compared to the D0 from a single exp fitting. If we look at the 2D0 values of the pIRIR150 signal in excel file, for majority of the grains the 2D0 values are smaller than 100 Gy, which are too small regarding the true dating limit of feldspar. I think that is one reason why the saturated grains (De>2D0) have such high proportions.

(R2) I suggest that the authors can either revise the D0 values with the single exp fitting, or add some arguments in the manuscript to explain why they preferred the exp+linear fitting. In the latter case, I think the authors should point out that the 'saturation' discussed in the manuscript is not the true saturation that we usually talk about. I would agree that the proportion of saturated grains is just a

defined criteria to act as a bleaching proxy, so it is not important whether the grains are truly saturated or not.

(R2) In lines 222-223: 'The percentage of saturated grains for the pIRIR-150 signal is constantly higher than for the IRSL-50 signal, which is an artefact of the high IRSL-50 fading rates - i.e., an athermal signal loss during burial.' I agree with this argument. The higher fading rate of IR50 will result in a lower proportion of saturated grains. But there should be a second reason: the 2D0 of pIRIR150 is generally lower than the IR50 (refer to the data in the supplementary excel file).

- (A) Thank you for your suggestion, we chose to use an exponential + linear fitting for both $D_e$ and $D_0$ estimations as a conservative estimate of the onset of saturation (see e.g. Joordens et al., 2015 in Nature), to keep our analysis consistent within this study and because we noticed in calculating the De values that a single saturating exponential function fitted our feldspar data only poorly. Oftentimes it is difficult to satisfyingly fit a single saturating exponential function to feldspar IRSL/pIRIR dose response data, and an exponential plus linear fit sometimes provides a better fitting alternative (Guralnik et al. 2015). This was also our motivation to choose exponential plus linear fit for the determination of De and corresponding $D_0$ from our IRSL/pIRIR dose response data. This approach will indeed tend to overestimate the amount of grains in saturation, however, bearing in mind the high amounts of grains in saturation for our samples the possible impact on the interpretation of the data will be very limited. Also, it should be kept in mind that we use relative trends of the amount of saturated grains for the interpretation of transport and depositional dynamics, which thus will not be biased by the selections of dose response fitting (as long as the same fitting is used for all samples). Therefore, we see no added value in re-fitting our dose response data and the recalculation of characteristic dose values.

- (A) Within our manuscript we intend to use saturated grains as a defined criterion to act as a (non)bleaching proxy to compare the degree of bleaching of our samples among each other. Furthermore, we are not quite sure what the reviewer defines as "true saturation" of feldspars. The onset of saturation is typically estimated using the $2D_0$ criteria (Wintle and Murray 2006, Rad. Meas.), which is at ~85% compared to the full saturation of the sensitivity corrected DRC. Especially on the single-grain level $2D_0$ values can be highly variable.

- (A) We adapted your suggestion to add that the $2D_0$ of pIRIR-150 is generally lower than the IRSL-50 (supplementary excel file) which also contributes to a lower proportion of saturated grains for the IRSL-50 signal, please see lines 236 and 237.

Suggestion 3:

(R2) It seems that the g-values of IR50 and pIRIR150 are only measured for one sample, and these g-values were used for fading correction for all samples. If this is right, I suggest the authors to state clearly which sample was used for g-value measurement (as well as for dose recovery) in the manuscript. Actually, I wish the authors can measure g-values for more samples, especially for the sample about 25 km away from the apex. There is a reduction in percentage of saturated grains at this position, and the authors attribute this deduction to the contribution of fluvially transported sediments from a tributary of the Seti River, which bleached more than the bedrock grains. However, the low percentage of saturated grains could also result from a source with higher fading rates. Comparison of fading rates for samples at different positions will help to test this hypothesis.

- (A) The g-values as stated for IRSL-50 and pIRIR-150 within this manuscript are an average g-value (per signal type) of g-values from 4 samples: NCL-7619109, -111, -122, and -123. Per sample we measured three multi-grain discs. This was not clear from the text in the first version of this manuscript. Therefore, we added text in lines 145-149, and appendix B to

clarify the steps we took to obtain these g-values. The 4 samples under study are spread along the mass movement body, and the results show that their g-values ± their uncertainty are more or less coinciding with each other. Based on these outcomes we chose to correct all samples for an average of the four g-values measured.

- (A) The laboratory fading test was performed on multi-grain discs using the approach outlined in Auclair et al. (2003) on samples NCL-7619109, -111, -122, and -123 (Table 1, column in the middle). The burial ages were corrected for the measured fading rates with the *calc_FadingCorr* function from the R Luminescence package (Kreutzer, 2023). The time between irradiation and prompt measurements was 255 seconds (tc). The g-value was normalized to 2 days (172800 seconds) (tc.g_value).

- (A) Based on the outcomes of our fading analysis, we argue that the lower percentage of saturated grains at a distance of 25 km from the apex doesn't result from a sediment showing higher fading rates, since all measured fading rates ± their uncertainty coincide. Sample PK-A02 (NCL-7619109) originates from a location downstream of the confluence of the Seti river and the Bijayapur Khola tributary and doesn't show a significant difference in terms of fading.

Suggestion 4:

(R2) When apply minimum age model, sigmab is always an important and controversial parameter. There is no information about this parameter in the main text. But from the supplementary R script, I saw that the authors used a sigmab of 0.3. I think it is worthy to present this value in the main text and add some discussion why this value is chosen.

- (A) We indeed used a sigma-b value of 0.3, this value is based on comparative literature (Smedley, 2014; Brill et al., 2018). We added a few lines (177-182) to support our choice.

Specific comments:

Over through the manuscript, the $D_e$, $D_r$, $D_0$, $L_x$, $T_x$, etc. should be written with subscript.

- (A) Check

Sometimes it is written 'infrared', sometimes it is written 'infra-red'. It is better to use 'infrared' throughout the manuscript.

- (A) Check

Line 21: 'Single-grain infrared stimulated luminescence', add '(IRSL)' behind 'luminescence'.

- (A) Check

Line 64: 'Especially infra-red stimulated luminescence'. Change 'infra-red' to 'infrared'. Add '(IRSL)' behind 'luminescence'

- (A) Check

Line 71: 'by distinguishing those grains which capture the "true burial age" from those which inherit a heterogeneously bleached signal'. Maybe it is better to say 'by distinguishing those grains which are fully bleached before deposition thus capturing the "true burial age" from those which inherit a heterogeneously bleached signal''

- (A) Agree

Line 128: 'grainsize' to 'grain size', 'HCl- solution' to 'HCl solution'

- (A) Check

Line 130: '2.50 and 2.56 g/cm3'. Is it 2.56 or 2.58?

- (A) It is 2.56 g/cm$^3$

Line 130: 'heavy density separation' to 'heavy liquid density separation'

- (A) Check

Line 131: 'adding a liquid with a density of 2.63 g/cm3'. What kind of heavy liquid is used? Better to state.

- (A) It is LST (Solution of lithium heteropolytungstates in water)

Line 131: 'HF-' to 'HF'; 'the K-feldspars' to 'the K-feldspar grains'

- (A) Check

Line 134: 'dried overnight at 105°C and then weighted to calculate the moisture content by Loss On Ignition (LOI)'. It think we cannot call it LOI. LOI is for other stuff such as the carbonate content measurement. The sample is heated inside a furnace with temperature over 900 °C.

- (A) Agree, reference to LOI method has been removed

Line 140: 'SAR' to 'single aliquot regenerative dose (SAR)'

- (A) Check

Line 142: Better to give the model of the solar simulator. Is it a Hönle SOL2?

- (A) Yes it is

Line 147: 'The newly obtained radiocarbon samples were calibrated with the IntCal20 curve within the OxCal online environment (Reimer et al., 2020).' Please refer to my suggestion above, and introduce the radiocarbon samples in the section of materials and methods.

- (A) We adapted this recommendation, please see the explanation paragraph below suggestion 1

Line 153: 'and/ or' to 'and/or'

- (A) Check

Line 177: 'feldspar minerals' to 'K-feldspar grains'

- (A) Check

Line 180: 'fading g-value' to 'fading rate (g-value)'

- (A) Check

Line 183: 'the overestimation ratio of the IRSL-50 and pIRIR150 age for each sample.' Add 's' behind 'ratio' and 'age'

- (A) Check

Line 184: 'The palaeodose derived from the bMAM dose estimate ranges from' to 'The palaeodoses derived from the bMAM dose estimate range from'

- (A) Check

Line 185: '34.0 ± 6.32 to 337 ± 53.8 Gy'. I can understand that the authors want to make the De and error have same effective numbers. But is seems to be more reasonable to be '34.0 ± 6.3 to 337 ± 54 Gy'

- (A) We chose to keep the same effective numbers throughout the manuscript

Line 189: 'pIRIR-150 data' to 'pIRIR-150 signal'

- (A) Check

Line 190: '205 ± 88.9', '136 ± 15.9'. Change 88.9 to 89 and 15.9 to 16. Add 'ka' behind '205 ± 89'

- (A) Check

Lime 208-209: 'Populations of grains can be homogeneously or heterogeneously bleached. Heterogeneously bleached populations contain zeroed grains as well as grains with remnant doses (Duller., 2008).' It seems strange to place these sentences here. Maybe it is better to put them in section 3.2. And change 'Duller., 2008' to 'Duller, 2008'.

- (A) Agree, added to section 3.2

Line 212-214: 'Table 4 lists the overdispersion, the number of accepted and saturated grains, and the calculated bleaching proxies of saturated, best-bleached, and other grains for all samples and both feldspar signals (for details see section 3.2). The data of the three bleaching proxies always sum up to 100 percent.'

In the abstract and most of the main text, the three bleaching proxies are: percentage of saturated grains, percentage of well-bleached grains, and the overdispersion. But here the authors said that the three proxies are the percentages of saturated, best-bleached, and other grains. Please check and make it consistent throughout the manuscript. Similar controversial statement occurred on Line 220.

- (A) The three bleaching proxies indeed are: percentage of saturated grains, percentage of well-bleached grains, and the overdispersion. We adapted the text as the reviewer suggested.

Line 278: 'e.g. Guralnik et al., 2015', add ',' after 'e.g.'

- (A) Check

Line 302. 'Our data of the three bleaching proxies (Fig. 8a and b) lack an overall trend with runout distance.' Fig 8 is only for the proportions of grains. What three bleaching proxies do the authors mean here? The OD is in Fig 7a.

- (A) We adapted the text, please see line 315.

Line 317: 'proxies best-bleached', add 'of' behind 'proxies'

- (A) Check

Line 329: 'Age overestimated by a few thousand years'. But for many samples, age is overestimated by tens of thousand years.

- ▪ (A) Changed to 'thousands of years'

Appendix A: Please indicate which sample is used for these measurements. Also, please indicate whether the g-value is normalized to g2d or not. If not, what is the tc?

- ▪ (A) Sample NCL-7619125 has been used for the residual and dose recovery measurements, this information is added to the appendix title. It is normalized, please see lines 145-149 for more information.

Figure 3:  The recycling ratio figure is strange. Why are there 2 recycling ratios for the IR50 signal (another black circle covered by the red triangle)? Also, it is better to state which sample was used for this figure. In Line 519, change 'paleodose error' to 'relative paleodose error'.

- ▪ (A) We adapted the figure

Figure 4. It looks odd that fig 4 for sampling appeared after fig 3. If the authors add the information of the radiocarbon samples in the 'materials and methods' section, then fig 4 can be moved before fig 3.

- ▪ (A) Figure 4 has been moved and is now part of figure 1 as inset g, which seems more logical. All other figures are accordingly renumbered.

Figure 5. I suggest to use the empty circles and triangles for the fading corrected IR50 and pIRIR150 ages. It is easier to distinguish from the faded ages.

- ▪ (A) We prefer not to use 'empty symbols' since a lot of the (non-)fading pIRIR-150 data overlaps, and then it can be hard to distinguish them from each other since the 'empty symbols' background can overlap with a filled symbol.

Figure 8: Line 541: 'three bleaching proxies, e.g. best-bleached grains in dark red, saturated grains in dark brown, and other grains in light brown'. Please refer to my comment regarding Line 212-214. Here the three bleaching proxies are the proportions of three kind of grains, without OD. It is not what the authors stated in the abstract and the other parts in the main text. Please make it consistent. Also, change 'e.g.' to 'i.e.,'.

- ▪ (A) Agreed, we changed this to make it consistent.

Line 545: Please change 'and/ or' to 'and'. 'or' is not appropriate here.

- ▪ (A) Check

Tables

Table 1, Line 551: 'Lx indicates the natural or regenerative OSL dose obtained' Please check this sentence.

- ▪ (A) We did check it, seems clear to us

It is better to use 'IR wash at 150˚ for 500 s' in the last step of the cycle in the protocol of fading test, rather than 'MG infrared stimulation at 150˚ for 500 s'.

- ▪ (A) Check

Table 2: It is better to add a column to include the 'distance from apex' as well.

- (A) Check

Table 3 caption: 'including the raw age and the calibrated age.' Please replace 'age' with 'ages'.

- (A) Check